# The Role of Protein Arginine Methylation as Post-Translational Modification on Actin Cytoskeletal Components in Neuronal Structure and Function

**DOI:** 10.3390/cells10051079

**Published:** 2021-05-01

**Authors:** Britta Qualmann, Michael M. Kessels

**Affiliations:** Institute of Biochemistry I, Jena University Hospital—Friedrich Schiller University Jena, Nonnenplan 2-4, 07743 Jena, Germany

**Keywords:** protein arginine methyltransferase (PRMT), post-translational modification, neuromorphogenesis, actin cytoskeleton, actin nucleation, actin nucleator Cobl, arginine methylation, neuronal structure, dendritic spines

## Abstract

The brain encompasses a complex network of neurons with exceptionally elaborated morphologies of their axonal (signal-sending) and dendritic (signal-receiving) parts. De novo actin filament formation is one of the major driving and steering forces for the development and plasticity of the neuronal arbor. Actin filament assembly and dynamics thus require tight temporal and spatial control. Such control is particularly effective at the level of regulating actin nucleation-promoting factors, as these are key components for filament formation. Arginine methylation represents an important post-translational regulatory mechanism that had previously been mainly associated with controlling nuclear processes. We will review and discuss emerging evidence from inhibitor studies and loss-of-function models for protein arginine methyltransferases (PRMTs), both in cells and whole organisms, that unveil that protein arginine methylation mediated by PRMTs represents an important regulatory mechanism in neuritic arbor formation, as well as in dendritic spine induction, maturation and plasticity. Recent results furthermore demonstrated that arginine methylation regulates actin cytosolic cytoskeletal components not only as indirect targets through additional signaling cascades, but can also directly control an actin nucleation-promoting factor shaping neuronal cells—a key process for the formation of neuronal networks in vertebrate brains.

## 1. Introduction

### 1.1. Protein Arginine Methylation

Post-translational modifications are important molecular mechanisms to fine-tune protein structure and function, but also to reversibly and thus temporarily modulate the activity and function of proteins. The methylation of arginine residues is catalyzed by protein arginine methyltransferases (PRMTs) resulting in ω-N^G^-monomethylated arginine (MMA), ω-N^G^, N^G^-asymmetric dimethylarginine (ADMA) and ω-N^G^, N’^G^-symmetric dimethylarginine (SDMA) (Figure 1) [1]. Protein arginine methylation is brought about by members of the protein arginine methyltransferase family, assigned to three subfamilies according to their modes of methylation (Figure 1 and Figure 2).

All subtypes catalyze the monomethylation of guanidino nitrogen atoms of arginine residues utilizing the methyl donor S-adenosylmethionine. Additional dimethylation of the monomethylated arginine is brought about solely by type I and II PRMTs, whereas the only known type III PRMT, PRMT7, solely exhibits monomethylation catalytic activity because of its distinctive structural characteristics. Type I (PRMT1, PRMT2, PRMT3 and PRMT4—also known as CARM1, PRMT6 and PRMT8) PRMT enzymes can additionally catalyze the formation of asymmetric dimethylarginine as an end-product, while type II PRMTs (PRMT5 and PRMT9) form in a second, subsequent step a symmetric dimethylarginine [1,2] (Figure 1). Two potential further PRMTs, PRMT10 and PRMT11, have not yet been validated but were predicted to display type II PRMT activity [3].

PRMTs exhibit different substrate specificities, whereby a certain preference for glycine–arginine-rich motifs has been noted for some family members (PRMT1, PRMT3 and PRMT6). The substrates of PRMTs known thus far mainly play a role in chromatin-mediated signaling and biogenesis and the maturation of ribosomes, in DNA damage responses and in mRNA processing and transport. PRMTs have been associated with various pathologies, particularly cancer, inflammation and immune responses [2,4]. The role of PRMTs in the development and prognosis of cancer pathologies is currently a major focus of basic and translational science.

Arginine methylation does not perturb the overall positive charge of the arginine guanidinium group. Instead, the addition of methyl groups reduces potential hydrogen bond interactions and furthermore significantly changes the shape and increases the size of the modified arginine. Thus, arginine methylation mainly results in steric effects as well as changes in hydrogen bond interactions [5].

### 1.2. PRMTs in the Central Nervous System

Almost 50 years ago, it was first recognized that arginine-methylated proteins are abundant in brain extracts [6]. During the following decades, PRMTs have been implicated in the pathogenesis of different neurological diseases, including amyotrophic lateral sclerosis, glioblastomas and Huntington’s disease [1]. In the central nervous system (CNS), they have been attributed to functions in cell maturation and differentiation but also neurodegeneration [2,7,8]. Particularly, PRMT1, PRMT4 and PRMT5 have been tightly associated with oligodendrocyte and astroglial maturation and differentiation, as well as axon myelination, and have been implicated in neurodegenerative, demyelinating disease, and multiple sclerosis [9]. PRMT1 is essential for the development of neurons, astrocytes and oligodendrocytes [8] and is required for CNS development at embryonic and perinatal stages. CNS-specific *Prmt1* knockout mice exhibited morphological abnormalities of the brain and died within 17 days after birth [7]. Several studies furthermore highlight functions of PRMT8 in neuronal development and function [9] as described in detail in Section 3.

### 1.3. Neuromorphogenesis Powered by Actin Cytoskeletal Forces

During development, vertebrate brains acquire a highly sophisticated and complex architecture of neuronal networks representing a prerequisite for proper brain function. On one hand, this requires highly regulated and coordinated migration of different types and cohorts of neurons generated from stem cells. On the other hand, once these neurons have reached their destination, this relies on the elaboration of the astonishing morphological intricacy that neurons acquire during their differentiation, shaping their signal-sending (axonal) and signal-receiving (dendritic) compartments.

Studies of cultured hippocampal neurons prepared from embryos or newborn pups of rodents have allowed for defining and recapitulating of different stages of polarity and morphology establishment [11,12]. Whereas the cells first are characterized by a relatively round shape with a non-polar distribution of lamellipodia, they soon start to extend cylindrical protrusions that contain a growth cone at their distal end. At this early stage, these protrusions—called neurites—still lack the molecular and structural characteristics of mature axonal and dendritic processes. One of these immature neurites is then selected to form the axon with a large growth cone and starts to elongate rapidly. Subsequently, dendrites grow out and build an elaborate, branched dendritic arbor. This process is controlled by internal and external signals resulting in a continuing induction, outgrowth and retraction of dendritic branches. Finally, neurons form synaptic contacts by synaptogenesis and thereby establish networks with other cells [11,12]. In contrast to primary neurons, in immortalized cell lines, such as mouse neuroblastoma Neuro2A cells or cells derived from a pheochromocytoma of the rat adrenal medulla (PC12), not all of these stages and developmental processes can be observed.

The developmental processes described above represent extensive changes of cell morphology during neuromorphogenesis. The arborization of neurons requires breaking the surface of a sphere—for axon or dendrite initiation—or of a cylinder—for branch induction. Both have to overcome the odds of membrane resistance. A major mechanism for shaping membranes is the organization and dynamics of the membrane-associated cytoskeleton. Forces that can result in protrusion can be produced by the dynamics of the cortical actin cytoskeleton that is associated with the plasma membrane. The initiation and establishment of new actin filaments at specific areas at the cell cortex can thereby be utilized to bring about the forces necessary for the induction of the distinct protrusive elements formed by neurons, the axon and the dendrites, axonal and dendritic branches and dendritic spines. Importantly, for all of these processes, a close temporal and spatial control of actin filament formation is indispensable.

The assembly of new actin filament is an energy-consuming process. Furthermore, cells need to maintain a considerable proportion of actin monomers to be able to react to inner and outer signals quickly and efficiently. The restriction is achieved due to the fact that the initial assembly of actin monomers into actin nuclei, onto which further actin monomers can then be added spontaneously, is kinetically not favored but represents the rate-limiting step in actin filament formation. Thus, efficient actin filament assembly requires cellular factors that actively overcome this kinetic hindrance—so-called nucleation-promoting factors or actin nucleators. These cytoskeletal components are thus prime targets for control mechanisms. Strategies for the required strict temporal and spatial regulation of these effectors or of key elements of bigger effector complexes include the release of intramolecular autoinhibition, dissociation of intermolecularly acting inhibitors and post-translational modifications [13,14].

The importance of actin polymerization for the formation of protrusive structures in neuromorphogenesis was observed decades ago, as capping actin filament ends with cytochalasin and thereby preventing actin filament elongation inhibited process formation [15,16]. Besides actin nucleators (see below), elongation-promoting factors, such as Ena/VASP proteins, are able to promote F-actin filament formation. In line with this finding, they were found to play a critical role in neuromorphogenesis [17,18].

Despite the variety of cellular functions relying on actin polymerization, only a limited number of actin nucleators have been described so far. These include the Arp2/3 complex, formins and the WH2 domain-based nucleators including Spire and cordon-bleu (Cobl) [12,19].

The Arp2/3 complex is composed of two actin-related proteins (Arp2 and 3) and five additional proteins, and generates branched actin filaments, because the complex binds to the sides of actin mother filaments. The required activation of the Arp2/3 complex can be brought about by members of the Wiskott–Aldrich syndrome protein (WASP) family including WASP, N-WASP, the Scar/WAVE proteins, WASH and WHAMM [13,20]. The Arp2/3 complex was found to be particularly important for the correct translocation of growth cones and axon development [12,21]. N-WASP-mediated Arp2/3 complex activation at the plasma membrane of neurons, which is critically important for axonal development, hereby relies on the recruitment and activation of N-WASP by lipid- and F-actin-binding adaptor proteins, including syndapin I and Abp1 [22,23], by Rho-type GTPases such as Cdc42 and by phosphatidylinositol-(4,5)-bisphosphate (PIP_2_) [12].

In contrast to the important function of the Arp2/3 complex in the correct development of axons, the formation and arborization of dendrites was demonstrated to critically rely on the actin nucleator Cobl [24]. Cobl belongs to a rather novel group of actin nucleators, the WH2 domain-containing actin nucleators. This class of proteins makes use of multiple WASP-homology 2 (WH2) domains, small motifs that can interact with actin monomers, or a combination of WH2 domains and actin filament-binding motifs to bring together actin nuclei sufficiently large for subsequent spontaneous polymerization [12]. In mammals, this group of proteins consists of Spire [25], cordon-bleu (Cobl) [24], leiomodin2 (Lmod-2) [26] and JMY [27]. Thus, cells employ distinct actin nucleators to mediate the complex remodeling processes underlying neuromorphogenesis.

Information processing in the brain critically relies on shaping morphologically distinct, compartmentalized synapses. The postsynaptic part of the majority of excitatory synapses is localized on dendritic spines. In higher brain functions, the morphological plasticity of dendritic spines is a key element. Dendritic spines can be classified according to their morphologies. Mushroom-shaped spines are characterized by large bulbous heads, thin spines are marked by elongated necks and small heads, stubby spines show no apparent spine neck and filopodia-like dendritic protrusions are long and thin and do not possess any postsynaptic density [28,29,30]. These distinct morphologies of dendritic spines are thought to correlate with their state of maturation and functional properties. Filopodia are regarded as immature dendritic protrusions due to the observation that they have a rather low abundance in the mature brain [31]. Whereas the rather dynamic thin spines are more transient, mushroom spines, which are characterized by an extended postsynaptic density, are correlated with higher synaptic strength and increased stability for information storage [30]. Alterations in synaptic activity are accompanied by changes in the morphology, length and number of dendritic spines. An active role of the actin cytoskeleton in membrane remodeling requires the targeting of actin nucleation machineries to postsynaptic membranes and their specific activation at distinct membranes and postsynaptic areas. Emerging evidence indicates that ample signaling pathways, which interconnect synaptic activity with spine remodeling, target local actin dynamics. Therefore, such distinct methods of regulation of the actin cytoskeleton, including post-translational modifications, are key for the induction, maturation and plasticity of dendritic spines and thereby are ultimately important for learning and memory [32,33].

## 2. Neuromorphogenesis Controlled by Protein Arginine Methylation

Early indications for an involvement of protein methylation came from pharmacological drug inhibition studies. In PC12 cells, general protein methylation inhibition by applying dihydroxycyclopentenyl adenine (DHCA) prevented nerve growth factor (NGF)-induced neurite outgrowth without influencing cell growth, NGF-induced survival or cell flattening. Removal of DHCA led to fast protein methylation of several proteins and simultaneous neurite outgrowth. The results thus indicated that NGF-regulated protein methylation plays a role in neurite outgrowth from PC12 cells [34].

As lysine residues in proteins can also be methylated, addressing the role of specifically arginine methylation as well as a putative involvement of PRMTs requires the application of more specific inhibitors. The compound arginine methyltransferase inhibitor 1 (AMI-1) has been reported to exhibit low micromolar-level inhibition for the tested PRMTs, PRMT1, PRMT3, PRMT4 and PRMT6, but is inactive against protein lysine methyltransferases (PKMTs). However, also for one of the PRMTs analyzed, PRMT5, no inhibition was observed [35,36,37]. Remarkably, treating cultures of hippocampal neurons at day in vitro 9 (DIV9), a comparatively late time point in dendritic arbor development, with AMI-1 at 2.5 µM for 3 days resulted in increased dendritic arborization, as seen in Sholl analyses and quantification of dendritic terminal points, and in increased dendritic outgrowth [38]. However, opposing effects were observed at earlier stages of dendritic development. Both generally inhibiting methylation by applying methylthioadenosine (MTA) [39] and adenosine 2ʹ,3ʹ dialdehyde (Adox) [40,41] as well as interfering specifically with arginine methylation by applying AMI-1, respectively, led to a significant reduction in both dendrite number and dendritic branching in hippocampal neurons incubated at DIV4 with the inhibitors for 48 h [42]. In line with these results are recent observations for more mature neurons (DIV14) further demonstrating that the inhibition of protein arginine methylation reduces dendritic complexity. In this study, a reduction in Sholl intersections after 72 h of treatment with Adox or AMI-1, respectively, was observed [43].

Additional studies have been conducted to investigate a specific involvement of certain distinct PRMTs (Figure 2) by transient, siRNA-mediated knockdown of the respective enzymes in neurons. These give some explanation for the apparently contradictory results of the inhibitor studies, as several PRMTs were identified that play a role in dendritogenesis. Evidence is emerging that they seem to have different distinct functions that might be relevant at specific stages of dendritic development and arborization, adding additional levels of complexity (Figure 3).

Knockdown of PRMT4/CARM1, but not PRMT1, from DIV9 to DIV14 in primary hippocampal neurons significantly increased the complexity of dendritic arborization. An increase was detected in the total dendritic branch tip number, the total dendritic branch length and in Sholl intersections [44] (Figure 3). These observations in more mature neurons are in line with some of the AMI-1 effects reported, where the authors observed likewise increased dendritic complexity [38]. The effects of PRMT4/CARM1 were supposed to be mediated via the methylation of HuD, an RNA-binding protein that regulates the stability of mRNAs, including the one of BDNF [38]. CARM1 can negatively regulate HuD activity and inhibit neuronal differentiation [45,46]. Since the methylated portion of HuD was decreased in NGF-treated PC12 cells, the authors suggested that downregulation of HuD methylation is a possible mechanism of how NGF may induce differentiation of PC12 cells [45].

In earlier studies in the neuroblastoma cell line Neuro2A, siRNA-mediated knockdown of PRMT1 was reported to reduce the number of neurite-bearing cells induced by serum deprivation [47]. The authors suggested that this effect was mediated by Btg2, a PRMT1 binding partner [48], since Btg2 knockdown phenocopied the effects on neurite outgrowth in Neuro2A cells [47]. Such apparently different effects of PRMT1 knockdown [44,47] might be attributed to differences in the time points of developmental state analyzed, but likewise to the different cellular systems, with hippocampal neurons representing a much more physiological system. A recent study provided supporting evidence that PRMT1-mediated functions are critically important for neuromorphogenesis in primary neurons [43]. The SCY1-like pseudokinase 1 (SCYL1), which interacts with γ2-COP to form COPI vesicles that regulate Golgi morphology, was identified as a substrate for PRMT1. SCYL1 arginine methylation was shown to be important for the interaction of SCYL1 with γ2-COP, and the siRNA-mediated knockdown of SCYL1 inhibited axonal outgrowth in Rat-1 cells (Figure 3). The inhibitory effect was rescued by siRNA-resistant SCYL1, but not a SCYL1 mutant, in which the arginine methylation site was mutated. Consistently, the inhibition of hippocampal neurons with AMI-1 and Adox suppressed axon outgrowth. The authors propose a model where SCYL1 arginine methylation by PRMT1 affects protein trafficking via Golgi morphology alteration and modulates axon and dendrite morphogenesis in neurons [43]. Whether or to what extent this involves cytoskeletal factors, or rather Rab family proteins, as suggested by the authors, will be an interesting future line of research.

Recent studies firmly established the role of PRMT2, which localized to the dendritic trees of neurons in the hippocampus and showed accumulations at dendritic growth cones in cultured hippocampal neurons, in dendritic arborization. Overexpression of PRMT2 resulted in a highly significantly elevated number of dendrites and dendritic branch points and this phenotype was dependent on arginine methylation [42]. Loss-of-function studies in developing hippocampal neurons demonstrated that PRMT2 indeed is crucial for dendritogenesis. Whereas the re-expression of an RNAi-insensitive mutant of PRMT2 fully rescued the reduction in dendrites and dendritic branch points brought about by the RNAi-mediated loss of PRMT2, re-expression of a version of PRMT2 with a mutated inactive catalytic domain was not able to rescue the PRMT2 loss-of-function phenotypes [42]. Thus, PRMT2 can promote dendritic arborization in an arginine methylation-dependent manner and this ability is crucial for dendritogenesis (Figure 3).

## 3. A Role of Protein Arginine Methylation in Dendritic Spine Formation and Maturation—Signaling to Actin Cytoskeletal Factors?

Evidence from inhibitor studies and loss-of-function models for PRMTs, both in cells and whole organisms, is emerging that protein arginine methylation mediated by PRMTs additionally represents a regulatory mechanism in dendritic spine induction, maturation and/or plasticity. Intriguingly, this potentially even involves actin cytoskeletal components, at least as indirect targets through additional signaling cascades.

PRMT4 (CARM1) was detected at postsynapses in hippocampal neurons, applying both immunocytochemistry and electron microscopy, and shown to be enriched in postsynaptic density fractions by Western blot analyses [44]. Functionally, RNAi-mediated knockdown of PRMT4/CARM1 in dissociated neurons (DIV9 to DIV14) resulted in increases in spine width and density and in a higher proportion of mushroom-type spines at the expense of filopodia-like and thin spines. PRMT4/CARM1 deficiency furthermore increased the number and size of clusters of the NMDA receptor subunit NR2B and the cluster size of the postsynaptic protein PSD-95 [44]. These results suggest that PRMT4/CARM1 plays a role in the formation and maturation of dendritic spines and postsynapses (Figure 3). Support for a role of protein arginine methylation in dendritic spine maturation in general was obtained by accompanying inhibitor studies with AMI-1 (treatment with 10 µM at DIV10 for 4 days), which phenocopied some of these phenotypes. The pharmacological intervention resulted in an increase in spine width but not spine density, an increased proportion of mushroom-type spines and more and larger clusters of NR2B and PSD-95 in comparison to controls [44].

A potential involvement of some of the other PRMTs is less clear. The knockdown of PRMT3 in hippocampal neurons did not cause an effect on spine density, neither alone nor in combination with BDNF. However, the authors report that the increase in “spine area” seen in control neurons upon treatment with BDNF is no longer seen under PRMT3 RNAi. This effect was attributed to the PRMT3 binding partner and substrate sp62, a component of the 40S ribosomal subunit, whose knockdown phenocopied the effect [49].

In *Prmt1* knockout mice, spine number and behavioral phenotypes analyzed were unaffected, but differences were observed in response to lipopolysaccharide (LPS) treatment [50]. One explanation for these observations may be that constitutive loss-of-function of PRMT1 can be compensated for in development but that acute, stress-induced rearrangements of dendritic spines may still require PRMT1.

The protein arginine methyltransferase PRMT8 exhibits several unique features among this family of post-translational modifiers (Figure 2). PRMT8 has a highly restricted tissue expression, and was described to localize specifically to neurons in the central nervous system [51,52,53]. Furthermore, only PRMT8 within the PRMT family has an N-terminal myristoylation site mediating membrane targeting (Figure 2) [51]. Furthermore, PRMT8 is a multifunctional protein. In addition to its arginine methyltransferase activity, PRMT8 can act as a phospholipase D that hydrolyzes phosphatidylcholine into phosphatidic acid and choline [51,54]. In brain development and function, phospholipase D enzymes have crucial functions [55]. The zebrafish PRMT8 ortholog was found to be important for embryonic and neural development [54,56]. *Prmt8* knockout mice showed abnormal motor behaviors and decreased choline and acetylcholine levels [54], as well as altered perineuronal network formation in the visual cortex and visual acuity [57]. PRMT8 conditional deletion in neurons in mice affected multiple features of synaptic function and plasticity. These included an increased evoked neurotransmitter release at Schaffer collateral-CA1 synapses, an almost 3-fold increase in mEPSC frequency—which was surprisingly not accompanied by any changes in synapse or dendritic spine density—and reduced long-term synaptic plasticity [58]. Furthermore, significant reductions in levels of the NMDA receptor subunit GluN2A and of eukaryotic initiation factors were detected. These alterations in synaptic function were not accompanied by detectable changes in brain or neuron morphology. Context-dependent fear learning was impaired but locomotor or anxiety-related behaviors were not altered in *Prmt8* conditional knockout mice [58], differing from the behavioral phenotypes described by Kim et al. [54] and Lo et al. [59] that reported reduced anxiety in open field and elevated plus maze paradigms.

Recently, a role for the arginine methylation activity of PRMT8 in dendritic spine maturation that involves signal cascades targeting neuronal actin dynamics has been reported [59] (Figure 3). shRNA-mediated knockdown of PRMT8, which localizes to postsynaptic sites overlapping with PSD-95 in primary hippocampal neurons, resulted in a decreased density of mushroom spines but increased filopodia density. Whereas wild-type RNAi-resistant PRMT8 rescued this effect, a mutant restricted to nuclear expression did not. The arginine methyltransferase catalytic activity but not the phospholipase D activity was necessary for this function, because the phenotype could only be rescued by a phospholipase-deficient but not a methyltransferase-deficient mutant [59]. This lack of functional importance of the phospholipase D activity was somewhat unexpected, because previous studies in PC12 cells had shown, in contrast, that a phospholipase D-activity-deficient PRMT8 mutant was unable to stimulate neurite branching and outgrowth in NGF-stimulated PC12 cells [54]. This apparent discrepancy might be explained by differences in the cellular system or in applying gain-of-function versus loss-of-function approaches, but may also point towards different requirements for distinct PRMT8 functions in neurite arbor compared to dendritic spine development and maturation. In vivo—addressed by in utero electroporation—PRMT8–shRNA increased the density of filopodia on secondary apical dendrites of hippocampal CA1 neurons at P21 without having a significant effect on mushroom-type spines, thus partially resembling the phenotypes observed in dissociated neurons upon acute PRMT8 loss-of-function. Furthermore, loss of PRMT8 in dissociated neurons increased the density of excitatory synapses on dendritic shafts, accompanied by a concomitant reduction of synapse density in dendritic spines [59]. In dissociated primary hippocampal neurons from *Prmt8* knockout mice, filopodia density was increased, while mushroom spine density was not significantly affected. In vivo, however, in apical dendrites of hippocampal CA1 neurons at 6 weeks, the density of neither filopodia nor mushroom spines was significantly changed, while, selectively, the length of mushroom spines was increased [59]. In a previous study, no significant differences were observed for spines in the hippocampal CA1 area in terms of density or proportions of the different spine classes between wild-type and *Prmt8* knockout mice at age 10–14 weeks [58]. Together, these observations may indicate compensatory mechanisms in vivo and with brain maturation.

Interestingly, deficiency for PRMT8 reduced the ratio of F-actin to G-actin and slowed F-actin recovery in dendritic spines in FRAP experiments. This regulation of the actin cytoskeleton was attributed to mediation via the Rac1–PAK1–cofilin pathway [58]. The actin depolymerizing factor cofilin represents an important regulatory target. Phosphorylation of serine 3 by LIM kinase is known to inhibit cofilin-mediated F-actin severing. This stabilization of actin filaments is thought to represent a molecular mechanism for increased filopodia abundance and spine length [60,61]. Indeed, increases in phosphorylation of cofilin, Rac1 activity and phosphorylation of PAK1 were observed in *Prmt8* knockout brain lysates, together with changes in translation initiation factors [59]. This led the authors to conclude that PRMT8 promotes dendritic spine maturation via the Rac1–PAK1–cofilin pathway controlled by translation initiation factors.

The RasGAP SH3 domain-binding protein 1 (G3BP1), which plays a role in regulating translation in stress granules, was previously reported to be arginine methylated [62,63] and to be a substrate of PRMT1, PRMT5 and PRMT8 [64]. Deficiency of G3BP1 in mice results in behavioral defects as well as abnormal synaptic plasticity and calcium homeostasis in neurons [65]. Although PRMT8 and G3BP1 showed only very low spatial overlap in dendritic spines, deficiency for G3BP1 phenocopied the effects of PRMT8 loss-of-function on spine maturation and F-actin turnover and further experiments demonstrated the importance of arginine methylation of G3BP1 and PAK signaling on these functions [59]. The authors therefore concluded that PRMT8-dependent G3BP1 arginine methylation regulates its binding to translation initiation factors and thus translation repression. They furthermore proposed that this represents a mechanism, which in turn controls the Rac1–PAK signaling pathway, and thereby also cofilin-mediated actin filament dynamics underlying proper dendritic spine maturation.

## 4. Direct Modulation of Actin by Arginine Methylation

Although a large variety of post-translational modifications of actin itself have been identified, including phosphorylation, acetylation, methylation, ADP-ribosylation, arginylation, oxidation and ubiquitinylation, the exact molecular mechanisms of action and the functional consequences of these covalent actin modifications are still rather poorly understood [33,66]. Most previous research has rather focused on the importance of actin-binding proteins in the regulation of actin organization and dynamics, whereas actin’s post-translational modifications have not been a major focus for understanding modes of actin regulation.

In neurons, post-translational modification of actin in the form of phosphorylation was reported to play a role in neuronal maturation, affecting actin filament turnover, dendritic spine morphology and synaptic plasticity [67]. Recently, Kumar et al. [68] described actin R256 monomethylation by PRMT5. Amino acid exchanges in the corresponding residue, R258, in human α-actin isoforms were reported to underlie human diseases [69]. In the human smooth muscle-specific actin isoform, SM α-actin, the exchange of R258C or R258Hs result in a predisposition to thoracic aortic aneurysms and dissections (TAAD), together with an early onset of ischemic strokes due to a Moyamoya-like cerebrovascular disease [70,71]. Notably, actin R256 methylation occurred in a rather organelle-specific manner in the nucleus, and studies in yeast pointed toward a role of this post-translational actin modification in transcription [68]. Thus, it will be of future interest to determine how these observations might be linked to the corresponding human disease mutants and their pathologies.

## 5. Arginine Methylation of Actin Cytoskeletal Effectors Controls Neuromorphogenesis

Recent work has unveiled that protein arginine methylation also directly targets an actin filament promoting factor and regulates its function in neuromorphogenesis. Arginine methylation of the actin nucleator Cobl represents an important method of controlling Cobl’s cytoskeletal properties and its crucial role in dendritic arborization (Figure 3) [42].

Cobl has been identified as a crucial cytoskeletal component for dendrite and dendritic branch formation [24]. Via functional and physical interconnection with the calcium sensor calmodulin and with syndapin I and Abp1, Cobl substantially influences neuronal morphology. The actin nucleator Cobl brings about transient and locally restricted F-actin accumulations observed prior to and during dendritic branch induction [72,73,74,75]. In addition, Cobl functions in forming specialized F-actin-rich structures in non-neuronal cells [76,77].

Interestingly, PRMT2 has been identified as a direct interaction partner for the actin nucleator Cobl [42]. Thus far, PRMT2 was mainly associated with functions in transcriptional regulation, apoptosis and cell cycle progression and was linked to inflammatory responses, cancer and obesity [2,4]. Functional studies in hippocampal neurons demonstrated the physiological relevance of the interaction of PRMT2 with Cobl. General inhibition of methylation as well as specifically blocking arginine methylation by AMI-1 abolished Cobl-mediated dendritic arbor formation. Cobl associated with PRMT2 via its N-terminus and this selectively promoted methylation of the C-terminal WH2 domain-containing, actin-nucleating part of Cobl [42].

Among the family of arginine protein methyltransferases, only PRMT2 comprises a Src Homology 3 (SH3) domain (Figure 2), and it was this additional domain by which PRMT2 formed stable protein complexes with Cobl. Consistently, the functions of Cobl in dendritogenesis required PRMT2, complex formation mediated by the PRMT2 SH3 domain and PRMT2’s catalytic activity, as shown in loss-of-function and corresponding rescue experiments [42].

Molecular mechanistic studies unveiled that arginine methylation controls Cobl’s actin binding abilities—the crucial prerequisite for Cobl-mediated actin filament formation [24]. Inhibiting protein arginine methylation by AMI-1 or specifically knocking down PRMT2 substantially decreased the ability of Cobl to associate with actin [42], which represents a molecular key requirement for Cobl-induced actin nucleation [24]. The second of the three WH2 domains of Cobl has the highest affinity for actin binding [24] and may thus represent the initial key in actin nucleation. Thus, this domain is predestined to be controlled and modulated by fast, reversible post-translational modification reactions. Indeed, arginine methylation was specifically detected in the second WH2 domain of Cobl. The two residues identified to be arginine-methylated by mass spectroscopy, R1226 and R1234, are both located in the α-helix of the WH2 domain [42], which has been reported to bind to actin within the cleft between the two actin subdomains 1 and 3 and thus to be in close contact with actin [78].

Thus, arginine methylation of Cobl was shown to represent a key requisite for Cobl’s actin binding and its functions in the formation of the specialized neuronal architecture underlying neuronal network formation [42].

## 6. Additional Cytoskeletal Targets for Neuronal Arginine Protein Methylation

The original view that microtubules are not present in dendritic spines was changed by studies revealing the invasion of microtubules itself, and additionally kinesin motor proteins from the dendritic shaft to dendritic spine heads [79,80]. A recent study interestingly correlated functional specificities of the three homologous kinesin I proteins (KIF5A, KIF5B and KIF5C) in vertebrates with their diverse C-termini, where arginine methylation was detected in KIF5B and KIF5C, but not KIF5A [81]. KIF5B knockdown in hippocampal neurons led to a reduction in mEPSC frequency and mushroom spine density with a corresponding increase in the other spine subtypes and filopodia. A mutant version of KIF5B, where the arginines in two RGG motifs in the C-terminus were exchanged to histidines, preventing arginine methylation, failed to rescue the reductions in mushroom spine density and mEPSC frequency. Analyses of conditional knockout mice, in which *Kif5b* was ablated only after birth to avoid lethality, revealed defects in dendritic spine morphogenesis, synaptic plasticity and memory formation, thereby confirming the functional importance of KIF5B in controlling excitatory synaptic plasticity [81]. It will be interesting in the future to unveil the underlying molecular mechanisms by defining specific cargos transported by KIF5B to dendritic spines that might include actin cytoskeletal effectors or regulators mediating dendritic spine morphogenesis, maintenance and plasticity.

## 7. Perspectives

Several aspects of arginine methylation represent urgent topics to be explored in the future. First, it will be key to identify further direct cytoskeletal targets. This will help to clarify whether also for cytosolic effector proteins arginine methylation is an important common mechanism of functional control. Subsequently, the molecular mechanistic details of the effect of arginine methylation on the functional properties of the cytoskeletal effectors need to be exploited and followed up by cellular analyses. Together, such studies will provide important new insights into the cytoskeletal functions of cells in general and into crucial aspects of cytoskeleton-driven development and plasticity of neuronal shape, compartmentalization and function in particular.

Furthermore, it is critical to reveal whether arginine methylation is a post-translational modification that is rather constitutive, or whether it is also temporally reversed and thus represents a dynamic post-translational modification. Yet, the existence of arginine demethylases is still controversial [2,5,82,83,84,85].

It will also be of interest to analyze putative crosstalk and interconnections between arginine methylation and additional post-translational modifications. As an example, Akt-mediated serine phosphorylation and arginine methylation of polyglutamine-expanded androgen receptors occur at the same consensus site and were reported to have opposing effects on neurotoxicity [86].

## Figures and Tables

**Figure 1 cells-10-01079-f001:**
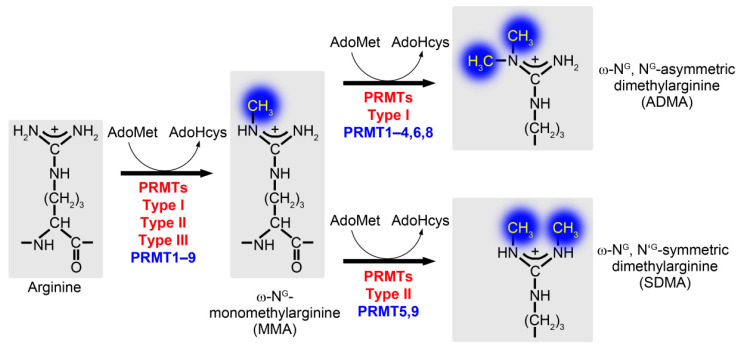
Protein Arginine Methylation by PRMTs. All three types of PRMTs (types I–III) are able to methylate one of the equivalent, terminal (ω) nitrogen atoms (ω-N^G^ and ω-N’^G^) using S-adenosylmethionine (SAM; AdoMet) as a methyl donor. The reaction leads to the generation of S-adenosylhomocystein (AdoHcys) and monomethylarginine (MMA). Type III PRMTs (PRMT7) exclusively catalyze this initial step. Type I PRMTs (PRMT1–4, 6 and 8) can in addition methylate the already monomethylated guanidine nitrogen atom of MMA further, leading to an asymmetric dimethylarginine (ADMA). In contrast, type II PRMTs (PRMT5 and PRMT9) methylate the second, thus far not methylated, terminal guanidine nitrogen atom of MMA (ω-N’^G^) and thereby give rise to a symmetric dimethylarginine (SDMA).

**Figure 2 cells-10-01079-f002:**
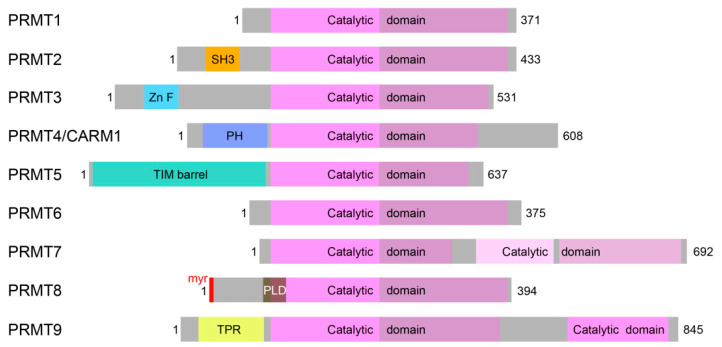
The family of human PRMTs. The catalytic core of PRMT1–9 is represented by the catalytic Rossman fold also known from other methyltransferases, such as protein lysine, DNA or RNA methyltransferases (light purple), as well as by the so-called β-barrel, which is involved in substrate recognition and PRMT dimerization (darker purple). In lighter shades are the less well studied C-terminal parts of PRMT7, with similarity to structural elements of the catalytic cores of other PRMTs, but without detected binding of the methyl donor S-adenosylmethionine. [10]. Other colors highlight additional domains found in the nine different PRMTs: orange, Src Homology 3 (SH3) domain of PRMT2; light blue, zinc finger (Zn F) of PRMT3; darker blue, Pleckstrin Homology (PH) domain of PRMT4/CARM1; turquoise, TIM barrel of PRMT5; red, N-terminal myristoylation (myr) site of PRMT8; yellow, tetratricopeptide repeats (TPR) of PRMT9. The region responsible for the additional phospholipase D activity (PLD) of PRMT8 is represented by a brown, transparent box, as it overlaps with the catalytic domain for arginine methylation.

**Figure 3 cells-10-01079-f003:**
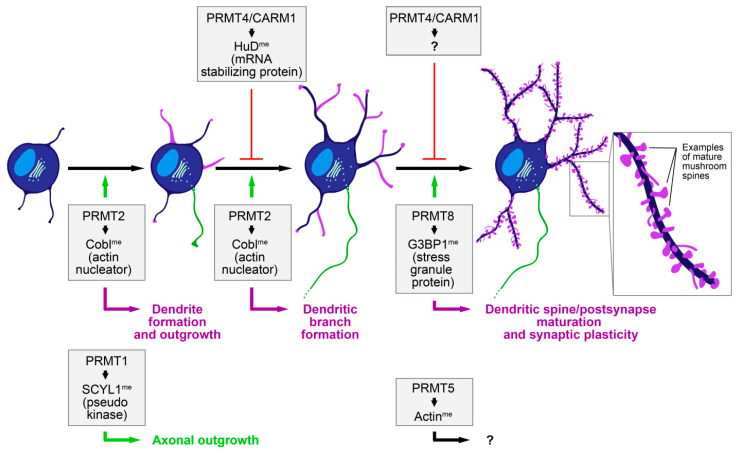
Neuronal development and the roles of individual PRMTs. Schematically depicted is a neuron undergoing neurite (left), dendrite (dark purple and magenta) and axon (green) formation as well as the outgrowth of these distinct compartments (second from left), dendritic branch formation giving rise to an extended and complex dendritic arbor (second from right) and dendritic spine formation and maturation (right). Morphological additions to dendrites in comparison to the previous stage are in magenta. Extensions of previously present structures are indicated by growth cones in magenta. Further structures present in the cell body are the nucleus (light blue) and the Golgi apparatus and additional secretory vesicles (light green). The information on PRMTs include arginine methylated (^me^) cellular targets (if identified) that are thought or have experimentally been demonstrated to bring about the respective neurodevelopmental function indicated. Whether a PRMT5-mediated methylation of actin plays a role in the nervous system is unknown and therefore not assigned to any neurodevelopmental stage or function.

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
