# Peer review of "The Role of Protein Arginine Methylation as Post-Translational Modification on Actin Cytoskeletal Components in Neuronal Structure and Function"

_cells, 2021, doi:10.3390/cells10051079_

Round 1
Reviewer 1 Report
This manuscript is a review of the currently-known roles PRMTs play in neuromorphogenesis, which is an emerging area of research for investigators interested in arginine methylation and brain development at cellular and molecular levels. The authors explain in detail required activities for dendritic and axonal formation involving their nucleating events, cytoskeletal components of these processes, and regulatory roles that PTMs can play with specific emphasis on arginine methylation. This review is needed to provide clarity and direction in this field, but I do think that some revisions will greatly improve this work.
Major revision:
The elaboration of PRMT activities and domain descriptions (section 1.1) and accompanying figures (Figures 1 and 2) can be found in just about most arginine methylation papers, but these details do not seem to be relevant or necessary to the rest of this manuscript (for example, text provided in lines 313 to 319 suffice and do not need a figure). At no time do the authors suggest that specific types of arginine methylation fulfill unique roles in regulating neuromorphogenesis, and only a small minority of PRMT domains are brought up, so these details in the only figures in the manuscript serve a hackneyed perfunctory role. In fact, given the level of detail in describing roles of cytoskeletal components in neuronal development, readers will greatly benefit by having figure(s) detailing neuronal development for axon and dendrite formation and actions taken by key players discussed in manuscript. Given the level of detail dedicated to this latter point, it is surprising that such figures are not in this manuscript.
Other revisions and edits:
Line 41: The PRMT family contains 9 bone fide family members and stating otherwise may be confusing to less-informed readers. The vast majority of papers in the PRMT field do not acknowledge PRMT10 and PRMT11 because there is insufficient evidence that they are actually PRMTs.
Section 1.2: It is surprising that PRMT8 is not introduced in this paragraph, especially given the full-throated discussion of PRMT8 later in the manuscript. There are plenty of references that detail its importance in the CNS (e.g., Penney et al (2017) describe PRMT8 and synaptic functions – not saying that this specific reference has to be cited, but PRMT8 is important enough to be revealed here).
Line 5: change pubs to pups.
Line 227-8: provide reference for PRMT7 catalytically inactive parts.
Line 250: insert period after [46].
Line 286: change trough to through.
Line 361: delete unnecessary hyphen.
Line 366: 10-14 days or months?
Line 383: What two proteins specifically?
Line 389: change actins to actin’s.
Line 475: sentence requires editing for clarity.
Line 486: More current references within the past couple of years are more suitable (“still is controversial”).
Reviewer 2 Report
This is an excellent review of the literature focused on Arginine methylation of proteins and its effect on neuronal structure and function. The two authors are highly regarded scientists in the field and treat the literature fairly. This is quite an extensive review but is constructed in a logical manner that makes it easy to read, for the most part. However there are a number of areas in the second half of the review that could use some editing to clarify the points that are being made. Specifically, some of the sentences are very long and confusing. Simplifying these sentences or breaking them into two or more sentences would make it easier to read and understand. Nevertheless, these changes should be easily addressable, as outlined below.
- Title: The title seems a bit wordy. Maybe “Arginine Methylation of the Actin Cytoskeleton affects Neuronal Structure and Function”?
Numbers refer to lines in the manuscript.
55- remove “to” in front of “play”
281 – “Potentially” weakens the title. Remove and replace with a question mark or just remove.
282-86 – Break into two sentences beginning at “potentially”
287-88 – Replace “applying both” with “with”. Also, is immunofluorescence IHC or ICC?
308 – Remove “essayed”.
355 – Should p21 be P21? Also a comma after “spines”
358 – Comma after “shafts”
359-363 – Break into two sentences at “however”.
362-63 – Remove “neither” and reword to “density was not significantly changed, while the length of mushroom spines was increased.”
366 – Should “age 10-14” be changed to P10-14 to be consistent with line 355?
366-67 – This sentence is unclear. To what is “respectively” referring? Why not just say “compensatory mechanisms with in vivo brain maturation”.
370 – Remove either “attributed” or “mediated” as together they are redundant.
375 – “kinase” should be removed after PAK1 since it is already in the name (K refers to kinase).
378 – Add “and is” after pathway. This makes the Rac1-PAK1-cofilin pathway separate, unless this pathway is itself controlled by translation initiation factors.
384 – Remove “that for”
387 – Remove “furthermore”.
386-390 – Sentence is very confusing and two “and thus” statements in it. Would benefit from breaking into two sentences as well.
391 – Remove “Cytoskeletal Factors” since this section is focused specifically on actin itself.
395 – Remove “on actin itself”. It is used earlier in the sentence.
396 – Reword “Research for a long time rather focused” to “Most previous research has focused”.
398 – “actins” should be actin’s or reword to “post-translational modification of actin”.
405 – “exchanges” could be reworded to “mutations”.
408 – “rather organelle-specific” could be reworded to “specifically”.
410 – Insert “to determine” between “interest” and “how”.
412-13 – Section title could be reworded to “Arginine methylation of Actin Cytoskeletal Effectors Controls Neuro-morphogenesis”
416 – Reword “mean in” to “way of”.
428 – Remove “Inhibition of methylation” as it is redundant.
430 – Remove “specifically” or move to before “promoted”.
436 – Comma after “activity”
438 – Switch “Molecular mechanistic” to “Mechanistic molecular”.
440-41 – Unclear why “specifically” is present. Do the authors mean either arginine methylation by AMI-1 OR knocking down PRMT2 “severely diminished Cobl’s ability to associate with actin?
475-76 – Sentence is unclear. Are the authors saying it will be key to identify further cytoskeletal targets of arginine methylation to determine if this posttranslational modification is an important mechanism of functional control?
477 - Switch “molecular mechanistic” to “mechanistic molecular” or just remove “mechanistic”.
484 – Remove “rather”.
487-88 – End sentence at “modifications” and start final sentence with “For example,”.
Round 2
Reviewer 1 Report
The author's of this manuscript have satisfactorily addressed comments from the previous review. There is one suggestion regarding the following point on a reference pertaining to PRMT7.
From Response:
"Point 5: Line 227-8: provide reference for PRMT7 catalytically inactive parts.
"Response 5: We considered the C terminal part of PRMT7 as catalytically inactive because important glutamates conserved in other PRMTs seem to be absent. To our knowledge there is, however, no literature that would explicitly experimentally prove that the C terminal part of PRMT7 indeed is catalytically inactive.
"This may anyway be difficult, as the catalytic activity of PRMT7 was reported as not very high and certainly only a fraction of the PRMT7 substrates have been identified up to today. We therefore modified the description in the legend of figure 2 and hope that this appropriately addresses the concerns indicated by the reviewer."
Reviewer's response: In Miranda et al. (2004) JBC 279: 22902-22907, the authors performed a UV cross linking experiment that showed the N-terminus capable of UV cross-linking to SAM, whereas the C-terminus was not able, suggesting that enzymatic activity comes from the N-terminus since SAM binding is a prerequisite for methylation activity.
Author Response
The additional reference suggested by the reviewer has been added. Thank you for this information on PRMT7 SAM binding.